# *Simarouba berteroana* Krug & Urb. Extracts and Fractions Possess Anthelmintic Activity Against Eggs and Larvae of Multidrug-Resistant *Haemonchus contortus*

**DOI:** 10.3390/vetsci12020090

**Published:** 2025-01-23

**Authors:** Marcos Javier Espino Ureña, Albert Katchborian-Neto, José Ribamar Garcez Neto, Francisco Flávio da Silva Lopes, Selene Maia de Morais, Vitor Eduardo Narciso dos Reis, Carmen Lúcia Cardoso, Lorena Mayana Beserra de Oliveira, Claudio Viegas Jr., Marcos José Marques, Wesley Lyeverton Correia Ribeiro

**Affiliations:** 1Department of Pathology and Parasitology, Institute of Biomedical Sciences, Federal University of Alfenas, 700 Gabriel Monteiro da Silva St., Alfenas 37130-001, MG, Brazil; mespino14@uasd.edu.do; 2Animal Production Center, Dominican Institute of Agricultural and Forestry Research, Duarte Ave., Km 24, Pedro Brand, 340 Parcel, Santo Domingo 10205, Dominican Republic; 3Faculty of Agronomic and Veterinary Sciences, Autonomous University of Santo Domingo, 1 Rogelio Roselle St., Engombe, Santo Domingo 10904, Dominican Republic; 4Institute of Chemistry, Federal University of Alfenas, 700 Gabriel Monteiro da Silva St., Alfenas 37130-001, MG, Brazil; 5Laboratory of Chemistry of Natural Products, State University of Ceará, 1700 Dr. Silas Munguba Av., Fortaleza 60714-903, CE, Brazil; jose.garcez@aluno.uece.br (J.R.G.N.); flavio.lopes@aluno.uece.br (F.F.d.S.L.); selene.morais@uece.br (S.M.d.M.); 6Doctoral Program in Biotechnology, Northeast Biotechnology Network, State University of Ceará, 1700 Dr. Silas Munguba Av., Fortaleza 60714-903, CE, Brazil; 7Postgraduate Program in Veterinary Sciences, Faculty of Veterinary, State University of Ceará, 1700 Dr. Silas Munguba Av., Fortaleza 60714-903, CE, Brazil; 8Group of Bioaffinity Chromatography and Natural Products, Department of Chemistry, Faculty of Philosophy, Science and Letters at Ribeirão Preto, University of São Paulo, Ribeirão Preto 14040-901, SP, Brazil; 9Laboratory of Research in Medicinal Chemistry—PeQuiM, Institute of Chemistry, Federal University of Alfenas, 2600 Jovino Fernandes Sales Ave., Alfenas 37133-840, MG, Brazil; claudio.viegas@unifal-mg.edu.br; 10Department of Physiology and Pharmacology, Faculty of Medicine, Federal University of Ceará, 1127 Coronel Nunes de Melo St., Rodolfo Teófilo, Fortaleza 60430-275, CE, Brazil

**Keywords:** endemic plant, exsheathment, phytotherapy, ovicidal, phenolic content, quassinoids, tannins

## Abstract

Gastrointestinal nematode infections pose a global threat to small ruminant production, worsened by the growing resistance to existing anthelmintic drugs. This study evaluated the in vitro anthelmintic activity and chemical profile of *Simarouba berteroana* extracts and fractions. High levels of phenolics, tannins, and flavonoids were quantified in crude extracts and medium- and high-polarity fractions; additionally, quassinoids, terpenoid quinones, phytosterol lipids, alkaloids, and naphthoquinones were annotated. The hydroalcoholic and *iso*-butanol fractions showed significant inhibitory effects on *Haemonchus contortus* egg hatching, and the extracts and fractions adhered to eggshells, leading to evisceration and cuticle detachment in larvae. Except for the hexane fraction, all tested extracts and fractions inhibited the exsheathment of third-stage larvae. These findings indicate *S. berteroana* as a promising source of natural compounds for developing new anthelmintic treatments.

## 1. Introduction

Parasitic diseases are a significant global challenge to animal health, posing a major obstacle to sustainable and more efficient food production [1]. Among them, gastrointestinal nematodes (GINs) represent one of the most critical health problems in small ruminant production systems, leading to substantial economic losses worldwide [2,3]. *Haemonchus contortus,* the most pathogenic nematode [4] and the most prevalent in tropical and subtropical regions of the world [5], is particularly concerning due to its hematophagous feeding habits, which can rapidly lead to severe anemia and death in affected flocks [6]. In addition, haemonchosis is of particular concern because *H. contortus* employs survival strategies such as larval developmental arrest at the fourth larval stage (L_4_) in the mucosa, a process known as hypobiosis, allowing it to endure unfavorable environmental conditions until favorable temperatures and humidity return [4]. The growing resistance of *H. contortus* to conventional anthelmintic drugs is a pressing concern [7]. Resistance has been documented across all major classes of anthelmintics, including benzimidazoles, macrocyclic lactones, imidazothiazoles, and even newer molecules like monepantel [8] and derquantel [9]. This escalating problem underscores the urgent need for novel anthelmintics with different mechanisms of action as alternatives to combat GINs effectively [6,10].

Phytotherapy has emerged as a promising alternative to synthetic antiparasitic and antimicrobial drugs, offering advantages such as effectiveness, greater accessibility in certain regions, lower costs, and reduced environmental impact [9,11,12]. Plant species of the family Simaroubaceae are well known for their medicinal properties and have been traditionally used for the treatment of malaria and parasitic infections [13,14,15]. A hallmark of this plant family is the widespread presence of quassinoids, a natural product chemical class derived from the triterpenoids [16], which exhibit a range of biological activities [13]. Species such as *Simarouba glauca* have demonstrated antiparasitic properties, particularly in the management of ectoparasites, with bioactive compounds such as alkaloids and phenolic compounds, including tannins and flavonoids that were already isolated in the leaves [17,18]. Nevertheless, quassinoids, including ailanthinone, glaucarubinone, and holacanthone, are recognized as the primary bioactive metabolites in Simaroubaceae plants, exhibiting potent antiparasitic effects [19]. Notably, aqueous and ethanolic extracts of *Picrolemma sprucei* (Simaroubaceae) stems and roots, as well as its isolated quassinoids, neosergeolide, and isobrucein B, have demonstrated significant in vitro anthelmintic activity, including lethality against the third-stage larvae of *H. contortus* [15].

In recent studies, Brazilian medicinal plants traditionally used as anthelmintic agents ranked Simaroubaceae as the fourth most frequently cited family, with *Simarouba versicolor* being the most commonly reported species [14]. Given the phytochemical similarities among species within the *Simarouba* genus, it is hypothesized that *S. berteroana* may exhibit comparable biological activities, including anthelmintic effects. Thus, *S. berteroana* Krug & Urb., commonly known as “Olivo”, is an endemic plant on the Caribbean Island Hispaniola that can grow up to 20 m in height (Appendix A) and is usually found in Pedernales, Dominican Republic [20]. Despite its traditional use in ethnomedicine, pharmacological and phytochemical studies are limited. However, previous research of Devkota et al. [21] isolated a series of compounds from *S. berteroana*, including eight canthine alkaloids, quassinoids, and one neolignane derivative [21], suggesting potential antiparasitic activity similar to other Simarouba species.

Addressing the need for alternative therapies for helminth control in small ruminants, the present study assesses the in vitro anthelmintic potential of the crude extracts and fractions of *S. berteroana* leaves, collected during two different phenological stages, against a multidrug-resistant isolate of *H. contortus*. To assess potential ultrastructural damage to the eggs and larvae of *H. contortus* induced by *S. berteroana*, fluorescence microscopy with propidium iodide was employed. In addition, liquid chromatography coupled with mass spectrometry (LC-MS) analyses were performed to evaluated the chemical composition of the extracts and fractions of *S. berteroana.* This research aims to establish *S. berteroana* as a viable source of natural anthelmintics, contributing to the development of sustainable alternatives for parasite control in small ruminant production.

## 2. Material and Methods

### 2.1. Collection and Management of Plant Material

The plant material was first collected in 22 February 2020 (at the end of the winter), in its natural habitat, in Pedernales province, Dominican Republic (17°50′55″ N, 71°20′11″ W), with 26 °C and a relative humidity of 75%. It was identified by a taxonomist, and a voucher specimen was deposited at the National Botanical Garden (JBSD) “Dr. Rafael M. Moscoso” under the code 11831. Approximately 20 kg of plant material, consisting of thin stems, leaves, and fruits, was obtained and stored in plastic bags. Then, the plant material was subsequently dried in air, protected from sunlight for 14 days, and ground into 2–3 mm particles. The powder material was stored in plastic cages. A second collection was carried out in the same place on 15 June 2021 (late spring period), with 25 °C and a relative humidity of 89%, resulting in 29.87 kg of fresh leaves and stems (no fruits). The plant material was treated in the same way as the first one.

### 2.2. Obtention of Extracts and Fractions

The crude extracts from each collection were obtained using a percolator system for maceration with 95% ethanol for 24 h, followed by another extraction round for 45 min. The crude extracts were then concentrated under reduced pressure at 45 °C, resulting in 52.8 g and 127.2 g of the 1st and 2nd collected material, respectively (~6.8% yield), which were stored in dark glass containers at −20 °C until use. The crude extracts (10 g) from the 1st (Sb1) and 2nd collections (Sb2) of *S. berteroana* were resuspended in a 7:3 MeOH/H_2_O mixture (100 mL) and submitted to a liquid–liquid partition with hexane, ethyl acetate, and *iso*-butanol (40 mL each). As a result, the hexane (HexFr), ethyl acetate (EtAcFr), *iso*-butanol (isobFr), and remaining hydroalcoholic (FrHaq) fractions were obtained. Subsequently, all these fractions were concentrated under vacuum (rotatory evaporator), furnishing the eight corresponding fractions as follows: Sb1—HexFr (0.17 g, 2.8%), EtAcFr, (1.77 g, 28.9%), isobFr (1.03 g, 16.9%); Sb2—HexFr (0.11 g, 2.4%), EtAcFr (1.93 g, 43.7%), isobFr (1.56 g, 35.2%), and HaqFr (0.83 g, 18.7%).

### 2.3. Spectrophotometric Analyses

#### 2.3.1. Total Phenolic Content

The total phenolic content was determined by spectroscopy in the visible range using the Folin–Ciocalteu method adapted from Sousa et al. [22]. Briefly, 7.5 mg of each extract was dissolved in MeOH and then transferred to a 25 mL volumetric flask, and the final volume was made up with MeOH. A 100 μL aliquot of this solution was shaken for 30 s with 500 μL of Folin–Ciocalteu, and then 6 mL of distilled water and 2 mL of 15% Na_2_CO_3_ were added. The mixture was shaken again for 1 min and made up to 10 mL with distilled H_2_O. After 2 h in the dark, the absorbance of the samples at 750 nm was determined in triplicate using a UV-Vis spectrophotometer. The quantification of phenolic compounds was determined using the analytical curve of gallic acid: y = 0.1277x + 0.0118, R^2^ = 0.9953. The results are expressed in mg of gallic acid equivalent per gram of extract or fraction (mg EAG/g).

#### 2.3.2. Total Tannin Content

Total tannins in the crude extract and fractions were quantified according to the Folin–Denis method [23] and expressed as tannic acid equivalents. Briefly, 5 mg of each crude extract was dissolved in 100 mL of distilled water, and then 1 mL of this solution was added to a test tube with 1 mL of Folin–Denis reagent. The mixture was then shaken and allowed to stand for 3 min. Finally, 1 mL of 8% Na_2_CO_3_ was added to each solution, stirred, and allowed to stand for 2 h. After this time, the absorbance was read on a spectrophotometer at 725 nm. The equation for the calibration curve, y = 0.0748x − 0.0067, R^2^ = 0.9995, was used to determine the total tannin content, where x is the absorbance value of the readings from the tubes of each extract. The results are expressed in mg of tannic acid equivalents per gram of extract or fraction (mg TAE/g).

#### 2.3.3. Total Flavonoid Content

The total flavonoid content was determined by spectrometry in the visible range using the AlCl_3_ method adapted from Funari and Ferro [24]. To quantify the total flavonoids in the samples, a solution was prepared by dissolving 20 mg of the extract/fraction sample in 10 mL of ethanol. An aliquot of 2 mL of this solution (concentration 2 mg/mL) was mixed with 1 mL of 2.5% AlCl_3_ solution, and the volume was completed to 25 mL with ethanol. After resting in the dark for 30 min, the absorbance of the sample was determined at 425 nm on the spectrophotometer. The absorbance of the solutions prepared with the standard was determined in triplicate (3 times). The quantitation of flavonoids was determined using an analytical curve for quercetin. The curve used was y = 0.0677, x = 0.0114, R^2^ = 0.9992, where x is the absorbance value of the readings from the tubes of each extract. The results are expressed as quercetin equivalents per gram of extract (mg EQ/g).

### 2.4. Liquid Chromatography Coupled to Mass Spectrometry (LC-MS) Analyses

The chemical composition of the extracts and fractions was analyzed using a Nexera XR HPLC system (Shimadzu, Kyoto, Japan) consisting of two LC 20AD pumps, a Sil-20A injector, a DGU-20A degasser, a CTO-20A oven, and a CBM-20A controller. The LC system was coupled to an AmaZon Speed Ion Trap (IT) mass spectrometer (Bruker Daltonics, Bremen, Germany) equipped with an electrospray ionization (ESI) source. The HPLC-IT/MS(n) was controlled by Bruker Compass Hystar software (version 4.5). The chromatographic separation was performed with a Kinetex^®^ C18 100 A column (50 mm × 4.6 mm, 2.6 μm particle size). The solvents used were water (A) and acetonitrile (B), both phases acidified with 0.1% formic acid. The analysis was conducted in gradient elution mode at a flow rate of 0.3 mL/min under the following conditions: 0–40 min: 5% to 100% B; 40–45 min: 100% B; 45–50 min: 5% B. The extracts and fractions were diluted in methanol to a concentration of 1.0 mg/mL, and 10 μL of each solution was injected in duplicate alongside blank samples under the described conditions. The ion trap mass spectrometer parameters were optimized according to the manufacturer’s recommendations (Bruker) for the ionization source, with a chosen flow of 0.3 m/min: capillary voltage—4.5 kV (positive mode); end plate offset—550 V; nebulizer gas pressure (N_2_) 40 psi; drying gas (N_2_); gas flow—9.0 L/min; ionization temperature—300 °C. Analyses were performed in negative and positive ionization modes, covering a mass range of 50 to 1000 *m*/*z*. The raw data containing peak area and Rt–*m*/*z* pairs were exported to the .mzML format and then imported into MZmine 3.8.2 https://mzmine.github.io/ (accessed on 15 December 2024) for data visualization and analyses. Putative ion annotations in the analyzed extracts and fractions were performed by matching the nominal mass of protonated ions (ESI+) with the online Dictionary of Natural Products (DNP) v.33.1 [25].

### 2.5. Animals

Two 5-month-old male crossbred sheep of the Blackbelly and Santa Inês breeds were used as donors of infective *H. contortus* larvae to perform in vitro tests. Each sheep, initially free of gastrointestinal parasite infection, was infected with approximately 6000 third-stage larvae (L_3_) of a multidrug-resistant isolate of *H. contortus*.

### 2.6. Haemonchus Contortus Isolate

For both in vitro experiments, sheep were artificially infected with an *H. contortus* isolate known as Kokstad (KOK), which is recognized as being multidrug-resistant (MDR) and showing resistance to benzimidazoles, levamisole, and macrocyclic lactones [26,27,28].

### 2.7. In Vitro Anthelmintic Tests

#### 2.7.1. Egg Hatch Test (EHT)

The EHT was performed according to Coles et al. [29]. First, the feces of the animal harboring *H. contortus* were collected directly from the rectum and processed according to Hubert and Kerboeuf [30] to obtain a suspension containing the parasite eggs. Briefly, 250 μL of the egg suspension containing approximately 100 eggs was incubated for 48 h at 25 °C with 250 μL of the extracts and fractions of *S. berteroana* at different concentrations (16, 8, 4, 3, 2.5, 2, 1.5, 1, 0.5, 0.25 mg/mL), each diluted in distilled water with 0.5% DMSO. This assay was performed with two controls: 0.5% DMSO as the negative control and 0.1 mg/mL of thiabendazole as the positive control. After incubation, drops of Lugol’s iodine solution were added to stop egg hatching, and eggs and first-stage larvae (L_1_) were counted under a light microscope (Opton^®^) in the 10× objective. At least three repetitions were performed with four replicates for each treatment and control.

#### 2.7.2. Optical Fluorescence Microscopy

In addition to light microscopic observations to count eggs and hatched 1st-stage larvae, three samples of each treatment at each concentration were also evaluated in an optical fluorescence microscope (Nikon^®^, Tokyo, Japan) in the 10× and 40× objective lens to verify or confirm the ultrastructural changes detected in the light microscope. Thus, fluorescence microscopy of *H. contortus* eggs helped to evaluate the changes induced by the treatments in morphology, viability of eggs, and larval formation. Propidium iodide was used to confirm disruptions of the eggshell, egg membranes, or cuticles of 1st-stage-hatched larvae. Samples were prepared similarly to EHT with some modifications. The time of incubation was approximately 24 h (25 °C) and, after this period, the samples were kept refrigerated (8 °C) until the microscopic evaluation within a maximum period of 8 h. The mean and highest concentration of each extract and fraction, as well as the negative and positive controls, were evaluated, which were the same as for the EHT. The eggs were stained with propidium iodide (Sigma-Aldrich^®^, Saint Louis, MI, USA) at a ratio of 5:50, that is, 5 μL of propidium iodide (PI) for 50 μL of egg solution in contact with the respective treatments. The samples were then evaluated with the Nikon fluorescence microscope and visualized at 40× objective at 561 nm to detect PI in contact with dead or damaged eggs and larvae.

#### 2.7.3. Larval Artificial Exsheathment Assay (LAEA)

To recover *H. contortus* third-stage larvae (L_3_), feces were collected directly from the rectum of an experimentally infected sheep and processed according to Roberts and O’Sullivan [31]. Larvae were stored at 4 °C. Twenty-four hours before the experiment, 2- to 3-month-old larvae were transferred to 25 °C. The viability and proportion of exsheathed larvae were checked under a light microscope (Opton^®^) at 10× (viability greater than 97%). The test was performed according to Alonso-Díaz et al. [32]. The extracts and fractions were diluted in phosphate buffer solution (PBS) and the concentration evaluated was 300 μg/mL. Approximately 1000 to 1300 ensheathed L_3_ were incubated for 3 h at 25 °C with each extract and fraction. The larvae were washed and centrifuged three times in PBS (pH 7.2). The L_3_ was then subjected to the process of artificial exsheathment by incubation in a solution of sodium hypochlorite (2% *w*/*v*) diluted 1:300 in PBS and divided into 7 aliquots.

The kinetics of larval exsheathment was monitored under the optical microscope (10×) for 1 h, and exsheathed larvae were counted at 0, 10, 20, 30, 40, 50, and 60 min after contact with the hypochlorite dilution. PBS was used as a negative control. Four replicates were performed for each extract and fraction. The negative control was carried out to examine changes in the proportion of exsheathed larvae as a function of time. Also, each extract and fraction at 300 μg/mL was pre-incubated overnight with 50 mg/mL polyvinylpolypyrrolidone (PVPP) (Sigma-Aldrich^®^) to evaluate the role of tannins present in the extracts and fractions, as PVPP is a tannin inhibitor by forming a complex. Therefore, after the addition of PVPP to each extract and fraction, the assay was performed as previously described, including PVPP and PBS as a second negative control. At least four replicates were performed for each extract, fraction, and control.

### 2.8. Statistical Analyses

#### 2.8.1. Spectrophotometric Data

The values of total phenolic compounds and tannin and flavonoid contents were determined with at least three replicates and subjected to descriptive statistics. The mean and standard deviation of the contents of these compounds were determined using GraphPad Prism software v.8.0.2. Data were analyzed by one-way ANOVA and Tukey’s test was used to assess statistical differences (*p* < 0.05) between the values.

#### 2.8.2. In Vitro Anthelmintic Evaluation

In the EHT, the percentage effect of crude extracts and fractions was obtained using the following formula: %E = [(number of eggs/number of eggs + number of hatched L_1_)] × 100. The data were subjected to the Shapiro–Wilk test for normality analysis (*p* < 0.05). Since the data passed this test, they were analyzed by an analysis of variance (one-way ANOVA) followed by a comparison with Tukey’s test (*p* < 0.05). The effective concentration for inhibiting 50% (IC_50_) and 90% (IC_90_) of the egg hatching was determined by probit regression in the IMB SPSS statistics^®^ 22 software for Windows (New York, NY, USA). Additionally, a one-way ANOVA followed by Tukey’s test (*p* < 0.05) was performed to compare the IC_50_ and IC_90_ values of each extract and fraction. In the LAEA, the effect of crude extracts and fractions of the effect on the percentage of 3rd-stage larvae sheathing were obtained using the formula:%E = [number of exsheathed L_3_/(number of sheathed L_3_ + number of exsheathed L_3_)] × 100

Descriptive statistics of the values were then performed and the means and standard errors of the exsheathing percentages were plotted using GraphPad Prism^®^ 8.0.2 software.

## 3. Results

### 3.1. Chemical Analyses

#### 3.1.1. Phenolic, Tannins and Flavonoid Content

We determined the total phenolic, tannin, and flavonoid contents of the crude extracts and their fractions, which was the first time this has been performed for the studied species (Table 1). However, the yield of the hexane fraction of the *S. berteroana* crude extract (first collection) was insufficient for allowing the determination of the content of these compounds.

The *iso-*butanol fraction (isobFr) showed the highest total phenolic content in both Sb1 and Sb2 collections, followed by EtAcFr. There were no statistical differences between them, and their contents were statistically higher than those of the crude extract. In addition, HalcFr showed a total phenolic content that was statistically similar to the crude extract contents, but it was higher (*p* < 0.05) in Sb1 compared to Sb2 (288.71 and 189.6 mg GAE/g, respectively). The total tannin content was quantified, and the fractions isobFr and EtAcFr were shown to have the higher concentrations (517.9 and 500.0 mg TAE/g, respectively), followed by the crude extract (495.8 mg TAE/g), which was statistically similar. However, HalcFr showed a significantly lower total tannin content (324.2 mg TAE/g), followed by HexFr (15.45 mg TAE/g). Similarly, the crude extracts and their EtAcFr and isobFr fractions showed higher concentrations of total flavonoids than the HalcFr fractions. On the other hand, the EtAcFr of Sb1 showed the highest flavonoid content (30.88 mg QE/g), which was statistically different from the other fractions (*p* < 0.05).

#### 3.1.2. Liquid Chromatography Coupled Mass Spectrometry (LC-MS) Analyses

The LC-MS analyses of the extracts and fractions of *S. berteroana* revealed a distinct chromatographic profile (Figure 1), with 35 compounds annotated by matching nominal mass with metabolites previously isolated in plant species from the family of Simaroubaceae as well as from the genus *Simarouba*. Among these, 22 matched quassinoids of various types, including glycosylated and quinolone quassinoids, in addition to 5 phytosterol lipids, 3 terpenoid quinones, 3 alkaloids of the glycosylated canthin-6-one group, and 1 naphthoquinone. As expected, considering that the two collections were carried out in different seasons, it was observed that both crude extracts differ in their qualitative and quantitative levels, which suggests that the relative quantity and type of metabolites may differ depending on the seasonality.

The quassinoid putatively annotated as glaucarubol-15-deoxy, 2-ketone (Figure 1, final ID: 3) was among the most intense compounds in the chromatograms of the two crude extracts (Sb1 and Sb2) and the hydroalcoholic fraction (Sb1-HalcFr and Sb2-HalcFr) (Table 2). The compound annotated as glaucarubol 15-glucopyranoside (final ID: 28) also exhibited the highest peak areas in both crude extracts. In addition, the annotated quassinoid glaucarubol-13,18-didehydro-2-ketone-(2-hydroxy-2-methylbutanoyl) (Final ID: 17) was among the most intense annotated peaks, with the highest intensity in the crude extract of Sb1. Likewise, the annotated naphthoquinone javanicin D (Final ID: 33) was detected with a higher intensity in the crude extract of Sb2. Other compounds annotated as the terpenoid quinone simaroubin B (final ID: 34) and the phytosterol lipid named tirucalla-7,24-dien-3-one (stigmasta-7,22-dien-3) (final ID: 9) were among the most intense annotated peaks in both HalcFr, being higher in Sb1. In summary, the LC-MS analyses and the cross-matching of previously isolated compounds in the family and genus allowed for the detection and annotation of several quassinoids as constituents of both plant collections’ crude extracts and hydroalcoholic fractions. Other compounds detected and annotated in the samples, along with their biological source, SMILES, and chemical structures, are provided in the Appendix A.

### 3.2. Anthelmintic Tests

#### 3.2.1. EHT

Our experimental results demonstrate the effectiveness of the extracts and fractions of *S. berteroana* in inhibiting *H. contortus* egg hatching. Except for hexane fractions, the crude extracts and their hydroalcoholic, *iso-*butanol, and ethyl acetate fractions were the most effective at the concentrations used (Table 3 and Table 4).

The values of inhibitory concentrations of 50% and 90% in the egg hatch test evidenced that Sb1-HalcFr and Sb2-isobFr of *S. berteroana* were the most potent fractions, showing even higher potencies than their respective crude extracts on the eggs of the Kokstad isolate of *H. contortus* (Table 5). However, their effects (IC_50_ and IC_90_ values) were statistically similar to those of ethyl acetate fractions and crude extracts. Notably, the potent effect observed for Sb1-HalcFr in the inhibition of larval hatching was decreased compared to the same fraction obtained from the second collection (Sb2). Additionally, hexane fractions of both crude extracts showed the weakest anthelmintic effects (*p* < 0.05), with Sb2-HexFr being statistically more potent than Sb1-HexFr (Table 5).

It is also worth noting that a significant percentage of eggs exposed to crude extracts and high- and low-polarity fractions exhibited the complete development of first-stage larvae inside, which failed to hatch when observed under the optical microscope. The percentage of larval-formed eggs (%LFE) ranged from ~40% at concentrations of 0.5 and 1 mg/mL to over 80% at 2.5 and 3 mg/mL (Figure 2B,C). Additionally, the adhesion of extract particles to the outer surface of the eggshell was evident (Figure 2B,E). Furthermore, at low to medium concentrations (0.5 and 1 mg/mL), some ultrastructural alterations were observed in hatched larvae, including adhesions of extract material to the cuticle (Figure 2J,K), detachment of the cuticle from internal subjacent tissues (Figure 2I), and cuticle rupture with evisceration in some larvae (Figure 2G,H). The number of larvae exhibiting these alterations was quantified relative to the total number of larvae observed, and the resulting values are shown in the Appendix A. The percentage of L_1_ with these alterations was higher (~50%) in both isobFr and HalcFr from both collections, particularly at concentrations of 0.5 and 1 mg/mL (Appendix A).

#### 3.2.2. Optical Fluorescence Microscopic Evaluation

Given the observation of morphological alterations in the first-stage larvae hatching from the eggs of *H. contortus*, as well as in the eggs themselves, a decision was made to proceed with the photographic documentation of these eggs and larvae using an optical fluorescence microscope. Images were captured in two distinct modes: a clear field mode, which did not utilize fluorescence, and a fluorescence mode with propidium iodide at a wavelength of 561 nm. Figure 3 illustrates the typical changes observed in *H. contortus* first-stage larvae exposed to the crude extract and fractions of *S. berteroana* at intermediate concentrations (0.5–1 mg/mL). Similarly, Figure 3 depicts the changes observed in eggs treated with the crude extract and fractions of *S. berteroana* at higher concentrations (2–4 mg/mL). The images were observed at 10× and 40× magnification.

#### 3.2.3. LAEA

It was observed that, at 300 μg/mL, the exsheathment of *H. contortus* larvae was inhibited or delayed for 60 min to the greatest extent with the crude extracts, ethyl acetate, and *iso*-butanol fractions, followed by the hydroalcoholic fractions (exhibiting less inhibition), and then the hexane fractions (exhibiting no inhibition) (Figure 4). No significant differences in the inhibition of *H. contortus* larval exsheathment were observed among crude extracts from Sb1/Sb2 or among fractions from the same or different collections. However, Sb1-HalcFr exhibited a more pronounced inhibitory effect than the same fraction from Sb2 and showed comparable effectiveness to that of the crude extracts, isobFr, and EtAcFr.

The addition of the tannin inhibitor PVPP (50 mg/mL) led to restoring the ability of the larvae to exsheath. This phenomenon was predominantly observed in the crude extracts and fractions isobFr and EtAcFr, with a lesser effect for HalcFr. In the case of the hexane fractions, the addition of PVPP did not result in a perceptible difference in the ability of the larvae to exsheath, as these fractions did not exhibit any inhibitory or delaying effects on the third stage of *H. contortus* larvae.

## 4. Discussion

Gastrointestinal nematode infections pose a particular threat to small ruminant production systems, primarily due to the pathogenicity of *H. contortus* [2], which is known for its lethality and its ability to rapidly develop resistance to existing anthelmintic treatments [4,7].

In this study, we aimed to evaluate the effect of extracts and fractions of *S. berteroana* obtained at two different moments: upon egg hatching and upon the inhibition of third-stage larvae exsheathment of *H. contortus*. The decision to fractionate the crude extracts was motivated by the need to understand how variations in chemical composition might influence their biological effects and to identify which constituents may contribute most to their observed anthelmintic activity. Indeed, it was observed that crude extracts and fractions of *S. berteroana* exhibited in vitro anthelmintic activity against *H. contortus* eggs and infective larvae. The most active fractions in inhibiting egg hatching were the hydroalcoholic fraction from the first collection (Sb1-HalcFr) and the *iso*-butanol fraction from the second collection (Sb2-isobFr).

Concerning phytochemical analyses, there is a lack of published studies on *S. berteroana* extracts and fractions in the literature. Phenolic compounds, such as tannins and flavonoids, are of medium polarity [33]. As expected, it was observed that *iso*-butanol and ethyl acetate fractions were majorly concentrated in phenolic, tannin, and flavonoid content (Table 1), and, by contrast, the hexane fractions exhibited minimal concentrations of these compounds. The phenolic content of hexane, ethyl acetate, ethanol, and aqueous extracts derived from *Simarouba glauca* was found to be 0.7, 6.5, 16.5, and 5.9 mgGAE/g, respectively [34]. Additionally, the flavonoid contents of these same extracts were 0.6, 2.8, 4.8, and 4.3 mgQE/g. Therefore, the phenolic and flavonoid contents of *S. berteroana* extracts and fractions were found to be higher than those quantified in *S. glauca* by Jose et al. [34]. In contrast, another study reported that the phenolic contents of *S. glauca* aqueous, hydroalcoholic, ethanol, and ethyl acetate extracts were 92.38, 90.55, 84.55, and 76.69 mg GAE/g, respectively [35], being notably higher than those observed for extracts and fractions of *S. berteroana*. These findings suggest that the content of the phenolic compound type may vary significantly between plant species within a family, and even within the same species, depending on the solvent used to obtain the extracts or fractions.

Additionally, it is essential to note that the polarity of phenolic compounds varies depending on their chemical structure and complexity or degree of polymerization [36]. A phytochemical study of extracts derived from *S. glauca* revealed that the aqueous extract exhibited the highest total phenolic content (402.4 μg GAE/mg), followed by the methanolic extract, and the ethanolic extract demonstrated the lowest content (200–260 μg GAE/mg) [18]. Similarly to our study, the hydroalcoholic fraction of the first collection of *S. berteroana* (Sb1-HalcFr) exhibited a higher phenolic content compared to its crude (ethanolic) extract, even though Sb2-HalcFr showed a lower phenolic content compared to its crude extract. In addition, the total flavonoid content of *S. glauca* in the ethanolic extract was observed to be lower in the Umesh [18] study when compared to the crude extracts of *S. berteroana* of this study (14.98 μg QE/mg vs. 22.7–27.5 μg QE/mg). However, the aqueous extracts of *S. glauca* exhibited a higher flavonoid content (16.96 μg QE/mg) compared to the hydroalcoholic fractions of *S. berteroana* (12.49 and 8.61 μg QE/mg). This suggests that the phenolic compounds in the extracts of *S. berteroana* exhibit greater structural complexity or a higher degree of polymerization compared to those in the extracts of *S. glauca.* Consequently, these compounds were more concentrated in the crude extracts as well as in the ethyl acetate and iso-butanol fractions of *S. berteroana* but were more retained in the aqueous extract of *S. glauca* [33].

Our study revealed that Sb1-HalcFr exhibited a phenolic content statistically comparable to that of the isobFr and EtAcFr from the same correspondent collection. However, Sb2-HalcFr exhibited a statistically lower phenolic content than that of Sb1, which, in turn, exhibited a statistically lower phenolic content than isobFr and EtAcFr. Additionally, the total tannin content was found to be statistically lower in Sb2-HalcFr compared to Sb1-Halc, while the flavonoid content was also lower but not statistically significant. These variations can be attributed to the known changes in the metabolism of almost all living organisms depending on climate and season, especially in the specialized metabolism of plants, leading to changes in diverse biochemical pathways and fluctuations in the production of different metabolites at different times of the year. In addition, it must be considered that the time of collection of plant material, along with the phenological development stage and any external stressors affecting the specimens at a particular moment, can also influence the metabolites present in their tissues [37,38]. Consequently, the concentration of specialized metabolites in the tissues of a given plant will probably fluctuate in response to changes in one or more of these factors over time. Concerning this matter, the *S. berteroana* specimen collected in February 2020 (first collection, Sb1) was observed to be in a state of flowering, with the presence of multiple fruits. However, the specimen collected in June 2021 (second collection, Sb2) was observed to be in a phenological stage devoid of flowers or fruits. Therefore, these phenological variations, in conjunction with the differences in the time of collection, may be the underlying causes of the observed discrepancies, particularly in the hydroalcoholic fractions, despite the minimal differences observed in the crude extracts and the other fractions.

Moreover, the LC-MS analyses of *S. berteroana* extracts and fractions revealed a diverse array of phytochemicals, including quassinoids, quinone terpenoids, lipid phytosterols, alkaloids, and one naphthoquinone. Out of the 35 compounds putatively annotated, quassinoids dominated with 22. Also, the four most intense annotated peaks in the chromatograms were also predominantly annotated as quassinoids, thus suggesting that quassinoids are major constituents in the analyzed extracts and fractions.

In line with the present findings, Devkota et al. [21] isolated and identified compounds using LC-MS and 1D and 2D-NMR analyses. These compounds included two quassinoids, eight canthin alkaloids, and one neo-lignan. Notably, one of the identified canthin alkaloids, canthin-6-one-9-methoxy-5-*O*-*β*-*D*-glucopyranoside, was identified for the first time.

Other quassinoids have been isolated and identified in other *Simarouba* species as well. For instance, malikalactone D (SKD) has been isolated from *Simarouba tulae*, which is endemic to Puerto Rico [39]. Additionally, glaucarubinone has been identified in *S. glauca* [40], and 2′-acetylglaucarubine and 13,18-dehydroglaucarubinone have been identified in *Simarouba amara* from Guyana [41]. The detection and annotation of glaucarubinone (final ID: 4) in the present study further support the presence of this compound within the genus. Furthermore, other classes of compounds, such as alkaloids, terpenoids, steroids, and flavonoids, have been identified in other species of the Simaroubaceae family [13,15,16]. These results affirm that the LC-MS analyses of *S. berteroana* extracts and fractions are consistent with the typical phytochemical profile of the family, reinforcing the potential of this species as a rich source of bioactive compounds.

*Simarouba versicolor* was considered as a prominent plant with nematicidal properties, having numerous reports of its use [14]. They also referenced *S. amara* and *Quassia amara* as plants with similar properties. Additionally, *Ailanthus excelsa* has been identified as a species with both nematicidal and anthelmintic properties, along with other reported attributes [42] and *Picrolemma sprucei* Hook. F [15]. The majority of studies confirming the anthelmintic activity of plant derivatives from the Simaroubaceae family have focused on the mortality of third-stage larvae and adult nematodes of various species.

Zamilpa et al. [43] assessed the in vitro anthelmintic effect of a hexane extract from *Castela tortuosa* (Simaroubaceae) on the mortality of third-stage larvae (L_3_) of *H. contortus*. The findings revealed that, at 20 mg/kg, the mortality rate reached 78.2% after 72 h. Additionally, the lethal concentration of 90% (LC_90_) was found to be 98.2 mg/mL at 24 h and 64 mg/mL at 72 h. A comparison of these results with those of our study reveals that, although L_3_ mortality was not evaluated, the LC_90_ for the inhibition of egg hatching by the hexane fractions of *S. berteroana* from the first and second collections was 49.4 mg/mL and 74.8 mg/mL, respectively, both of which are lower than those determined by Zamilpa et al. [43]. It could be attributed that hexane extracts could act better on L_3_ mortality while medium- and high-polarity extracts are more potent for inhibiting egg hatching. The aqueous and ethanolic (95%) extracts from the leaves of *Samadera indica* (Simaroubaceae) demonstrated a concentration-dependent anthelmintic effect (10, 25, and 50 mg/mL) on two types of avian nematodes, inducing paralysis and death. The anthelmintic effect was observed in both *Raillietina spiralis* (a flatworm) and *Ascaris galli* (a roundworm) within a shorter time frame compared to Piperazine (10 mg/mL), which was used as a positive control [44]. Additionally, the authors attributed the observed anthelmintic activity to the phenolic compounds, such as tannins and flavonoids, present in the plant extracts. Consistent with the findings of Harindran and Rajalakshmi [44], *S. berteroana* extracts and fractions demonstrated a concentration-dependent anthelmintic effect against *H. contortus* in the egg hatch test, albeit at lower concentrations (0.25–16 mg/mL).

The decoction of *Spigelia anthelmia* was also evaluated in the egg hatch test using the *H. contortus* Kokstad isolate, exhibiting 90% and 99.3% inhibition of egg hatching at 5 and 10 mg/mL, respectively [45]. Additionally, the *S. anthelmia* decoction exhibited a phenolic content of 96.6 mgGAE/g of extract. A comparison of these results with those from our study reveals that the phenolic compounds in the crude extracts and fractions of *S. berteroana* were present at higher concentrations, with the exception of the hexane fractions. Furthermore, the egg hatch was inhibited by 90% and 100% at lower concentrations (2.5–3 mg/mL), except for the hexane fractions, showing the correlation between this ovicidal effect and the content of phenolic compounds, irrespective of the plant species from which the extracts were obtained.

Cortes-Morales et al. [46] evaluated the crude extract and its ethyl acetate and hydroalcoholic fractions of *Brongniartia montalvoana* in the inhibition of egg hatching of *H. contortus*. The crude extract and ethyl acetate fraction were shown to inhibit 99.7% and 99.2% of egg hatching at concentrations of 6 mg/mL and 0.8 mg/mL, respectively, while the hydroalcoholic fraction showed no effect at 6 mg/mL. These results partially align with our study, in which the ethyl acetate fraction (second collection) inhibited 99.1% of egg hatching at 2 mg/mL, outperforming both the crude extract (70.2% inhibition) and the hydroalcoholic fraction (60% inhibition) at the same concentration. However, the results differed for the first collection, where both the crude extract and ethyl acetate fraction inhibited approximately 76% of hatching at 2 mg/mL while the hydroalcoholic fraction was more potent, showing 100% inhibition at the same concentration. Therefore, the ovicidal effects of extracts and fractions are directly related to the concentration of phytochemicals, which is influenced not only by the plant species and type of solvent used for extraction [46,47] but also by the timing and conditions of plant collection.

High-resolution microscopy techniques, including scanning electron microscopy, transmission electron microscopy, confocal laser scanning microscopy, and fluorescence microscopy, are employed to observe specific changes in nematodes at various life stages (adult, larval, or egg) to gain insight into the direct effects or modes of action of plant derivatives and their phytoconstituents as nematicidal agents [46,48,49,50]. In this study, fluorescence microscopy using PI was employed to ascertain whether the constituents of the *S. berteroana* extracts and fractions caused any disruption or rupture of the eggshell and/or membranes of *H. contortus* eggs as part of their mode of action to inhibit egg hatching. Additionally, the objective was to determine whether the damage observed in hatched first-stage larvae resulted from any disruption of or alteration in the cuticle. The rationale for utilizing PI as a DNA staining reagent is that it can only penetrate cells when the cell membrane is injured or ruptured. Once inside the cell, PI intercalates into double-stranded DNA, resulting in red fluorescence [51]. It was observed that eggs in contact with *S. berteroana* crude extracts and fractions, especially at higher concentrations (≥1 mg/mL), contained fully developed larvae inside. Additionally, these eggs were more oval in shape and larger in size due to the effects of larval development. The first-stage larvae within the eggs exhibited some morphological changes, such as wrinkling. Upon examination of the eggs using fluorescence microscopy with PI, it was confirmed that the eggshell and membranes were damaged or disrupted, as evidenced by the red–orange staining of the egg interior and the formed larvae inside.

Furthermore, the morphological changes observed in hatched larvae, including larval evisceration and cuticle separation or detachment from the larval body, were examined under fluorescence microscopy. It was also noted that PI penetrated the larval body, indicating that these larvae were dead and that their cuticle and membrane integrity had been compromised. Moreover, eggs and first-stage larvae exposed to 0.5% DMSO (negative control) following the addition of PI did not exhibit orange staining, indicating the maintenance of membrane and cuticle integrity.

A previous study corroborated the impact of the essential oil of *Lippia dominguensis* on eggs and first-stage larvae of *H. contortus* using confocal microscopy. At a concentration of 2 mg/mL, the eggs exhibited red–orange staining with PI when a Live/Dead™ (Invitrogen™, Waltham, MA, USA) stain was employed [48]. Additionally, it was observed that the majority of the eggs contained fully developed first-stage larvae, a finding that is similarly evident in this study with *S. berteroana* extracts and fractions. Therefore, it was plausible to suggest that the ultrastructural alterations were caused by the transcuticular diffusion of the essential oil constituents, which, in turn, caused damage or changes in the permeability of the eggshell membrane, potentially inhibiting the larvae from hatching.

Confocal laser scanning microscopy was employed to observe the effects of *B. montalvoana* crude extract (6 mg/mL), an ethyl acetate fraction, and a subfraction (800 μg/mL) on *H. contortus* in the egg hatch test conducted by Cortes-Morales et al. [46]. The crude extract, fraction, and subfraction caused deformations in the eggs, with colocalization (interaction) observed between the green autofluorescence from the larvae and the blue fluorescence from the extract’s compounds (flavonoids, coumarins, and hydroxycinnamic acid derivatives) [46]. This suggests an interaction between the compounds and the larvae within the eggs. These authors hypothesized that the ethyl acetate fraction and its subfraction contain compounds capable of penetrating the egg’s triple-layer membrane and binding to the larvae, resulting in structural damage and death, although no damage to the eggshell was observed. These findings were consistent with another study that evaluated the ovicidal activity of isokaempferide from *Baccharis conferta* on *H. contortus* larvae [47].

Several mechanisms have been proposed to explain how phytochemicals, such as phenolic compounds, exert their inhibitory effects on egg hatching. Phenolic compounds can disrupt various key biological processes essential for larval development and their subsequent emergence from the eggshell. Phenolics may bind to the lipoproteins of the eggshell membranes, altering their permeability, or form a coating around the egg, which mechanically prevents the larvae from breaking through the eggshell [52,53]. Additionally, phenolic oxidation or hydrolysis products can bind to or interact with eggs through noncovalent or covalent interactions, leading to damage [52]. Another mechanism by which phenolics, such as tannins, inhibit egg hatching is through binding to enzymes (lipase, leucine aminopeptidase, and metalloproteases) and molecules essential for eggshell degradation [54]. This binding inactivates or reduces the production of these enzymes and molecules, thereby preventing the breakdown of the eggshell [54]. Furthermore, low-molecular-weight phenolics (400–500 Da) can penetrate the eggshell membranes without disrupting them, binding to the larvae and impairing their normal development by interfering with vital functions [46,53,55]. These compounds may also reduce normal larval motility, which is essential for breaking through the eggshell membranes and hatching [56].

The unusual shapes of the hatched larvae observed in this study, including evisceration, split-headed larvae, cuticle detachment from the body, and their inability to hatch properly, are likely due to the binding of phenolic compounds to the eggshells and larval proteins. This binding impairs egg hatching or results in larvae with morphological alterations, as observed at lower and middle concentrations of the *S. berteroana* extracts and fractions. At higher concentrations, the effect may be more related to the coating of the extracts and fraction compounds on the eggshell, which may mechanically inhibit egg hatching and bind to membrane proteins, disrupting their proper degradation.

The nematicidal activity of 38 quassinoids isolated from Simaroubaceae was tested against a species of *Diplogastridae* (nematode) [57]. Samaderines B and E, along with brucein D, demonstrated greater efficacy in killing the nematode than the synthetic anthelmintics albendazole, thiabendazole, and avermectin. Watanabe et al. [57] observed morphological changes in the larvae (coil or S-shaped curvature) after quassinoid exposure, similar to the effects seen with picrotoxinin, suggesting a similar mode of action as a GABA receptor Cl-channel complex antagonist. These authors also proposed that quassinoids exert their effects by inhibiting nematode protein synthesis. Therefore, it can be hypothesized that the quassinoids present in *S. berteroana* extracts and fractions, many of which possess a molecular weight below 500 Da, may act by penetrating the eggshell membranes and subsequently interacting with the developing larvae. This interaction could potentially disrupt normal motility by affecting the neuromuscular system or inhibiting proper protein synthesis, ultimately contributing to the inhibition of egg hatching. It can be reasonably deduced that the diverse phytoconstituents of *S. berteroana* acted in a synergistic or additive manner for inhibiting the egg hatch, through the aforementioned mechanisms of action, given the wide range of compounds present in *S. berteroana* extracts and fractions.

Phenolic compounds, including tannins and flavonoids, have been shown to affect several pivotal biological processes within the parasitic nematode life cycle, such as the establishment of infective L3 [53,58,59], by inhibiting the process of exsheathment [60,61]. Furthermore, phenolic compounds have been observed to impair the proper motility of L_3_ [49,52,56]. The mechanism behind the inhibition or delay of L_3_ exsheathment is related to the ability of tannins and other phenolics to bind to proteins and other macromolecules, resulting in changes to their physical and chemical properties [62]. This is particularly relevant given that the nematode cuticle is known to be a structure rich in proline and hydroxyproline [60,62].

Greiffer et al. [49] investigated the inhibitory effects of phytoconstituents from *Combretum mucronatum* extract on the exsheathment of *Caenorhabditis elegans* larvae using differential interference contrast microscopy, fluorescence microscopy, and atomic force microscopy. Their findings demonstrated that the cuticle serves as the primary binding site for tannins and that structural disruption of the cuticle impaired molting at all larval stages. They further noted that tannin–cuticle binding increased cuticle rigidity without affecting the underlying tissues. Additionally, while tannins did not inhibit the synthesis of new sheaths and cuticles, they interfered with the removal of the old cuticle, altered sheath flexibility, impaired motility, and caused muscle filament alterations, ultimately leading to larval death [49].

To the best of our knowledge, this is the first study to evaluate the inhibition of *H. contortus* third-stage larvae exsheathment by *S. berteroana* derivatives. It was observed that, except for hexane fractions, the crude extracts and fractions of *S. berteroana* inhibited or delayed the shedding of *H. contortus* L_3_ at a concentration of 0.3 mg/mL. However, the hydroalcoholic fraction from the second collection exhibited a somewhat less pronounced effect between 50 and 60 min (Figure 4), which may be related to the lower content of tannins, phenolics, and flavonoids in this fraction, as supported by the literature [60,63]. These findings are consistent with those reported by Oliveira et al. [64], who evaluated the exsheathment of *H. contortus* L_3_ (Kokstad isolate) using leaf extracts from tannin-rich plants (*Anadenanthera colubrina*, *Leucaena leucocephala*, and *Mimosa tenuiflora*). The authors observed that larval exsheathment was inhibited at a concentration of 0.31 mg/mL. Additionally, they reported that the phenolic content of the evaluated extracts ranged from 48.2 to 147.9 mg GAE/g, with a tannin content between 30.1 and 138.85 mg GAE/g. Similarly, Macedo et al. [65] investigated the effects of decoctions of *Lantana camara*, *Alpinia zerumbet*, *Mentha villosa*, and *Tagetes minuta* on the larval exsheathment of *H. contortus* (Kokstad isolate) by quantifying total phenolic and tannin content. They found that the *A. zerumbet* and *M. villosa* decoctions exhibited a higher phenolic content (116.2 and 117.7 mg GAE/g, respectively) and tannin content (107.1 and 87.5 mg GAE/g) compared to the other decoctions. Consequently, these decoctions also exhibited complete inhibition of larval exsheathment at a concentration of 0.31 mg/mL, while the other decoctions achieved the same effect only at 0.62 mg/mL.

A comparison between the results of the aforementioned studies and those of the present study indicates that the plant extracts evaluated in the previous studies had a higher phenolic content than the crude extracts and fractions of *S. berteroana*. Nonetheless, the impact on the larval exsheathment of the same *H. contortus* isolate was found to be quite similar to that observed in the present study, except for the hexane fractions, which did not show inhibition of larval exsheathment, and also showed the lowest phenolic contents.

Klongsiriwet et al. [66] demonstrated that the combination of condensed tannins and flavonoids, such as quercetin and luteolin, enhanced anthelmintic activity and exhibited a synergistic effect on larval exsheathment inhibition, even at lower concentrations of procyanidin and prodelphinidin tannins. In our study, the inhibition of larval exsheathment with *S. berteroana* extracts and fractions is likely due to the synergistic action of various phenolic compounds, including tannins and flavonoids. The action of phenolic compounds in plant extracts on larval exsheathment can be further confirmed by the addition of a substance capable of inhibiting tannins and other phenolic compounds through complexation, such as polyvinylpolypyrrolidone (PVPP) [32]. In such cases, the larvae’s typical ability to exsheath would be restored. In this study, the inhibitory effects on larval exsheathment observed with *S. globosa* crude extracts and *iso*-butanol, ethyl acetate, and hydroalcoholic fractions were not observed when PVPP was added to the larval solution. Instead, larval exsheathment was similar to that of the control groups treated with PBS and PBS + PVPP. Therefore, these results confirmed that the inhibition of the exsheathment of the L_3_ was due to the action of the phenolic compounds presents in these extracts and fractions, which were inactivated or complexed by PVPP. These findings are consistent with previous studies [64,65,67]. Regarding quassinoids, it appears that they exerted a minimal impact on larval exsheathment, as the addition of PVPP (which inhibited the phenolic action) resulted in the complete restoration of the ability of *H. contortus* L_3_ to exsheath. Further studies are required to gain a deeper understanding of the anthelmintic effects of *S. berteroana* quassinoids on larval exsheathment, as well as to elucidate the mechanisms of action underlying their effects on various stages of the *H. contortus* life cycle.

It is important to consider that, despite the anthelmintic potential of phenolic compounds, they have also been observed to exhibit certain undesirable effects, including a reduction in nutrient digestibility (specially proteins) due to the formation of complexes between macromolecules and tannin structures [68]. Additionally, these compounds have been shown to decreased diet palatability and to reduce feed intake [69]. In light of this, it is recommended that the potential anti-nutritional effects of *S. berteroana* extracts and fractions be evaluated alongside their in vivo anthelmintic efficacy.

The relatively lower concentrations of *S. berteroana* extracts and fractions required to achieve anthelmintic effects on *H. contortus* eggs and larvae, as observed in this study, suggest a high potential for anthelmintic efficacy in sheep and goats. This is particularly evident when compared to results reported in other studies that used similar or higher concentrations of plant derivatives [43,44,45,46]. Furthermore, future studies should aim toward the optimization of plant collection and perform a detailed chemical profile across different harvests to identify the periods and conditions under which the plant’s anthelmintic activity is maximized.

## 5. Conclusions

This study provides the first evidence of the in vitro anthelmintic activity of *S. berteroana* extracts and fractions, demonstrating their potential in controlling *H. contortus.* The hydroalcoholic, *iso*-butanol, and ethyl acetate fractions, along with the crude extracts, exhibited significant efficacy by inhibiting larval hatching and delaying the exsheathment of infective larvae. These effects are likely associated with the high levels of phytochemicals present, including phenolic compounds, such as tannins and flavonoids, and also quassinoids, in addition to phytosterol lipids. The phytoconstituents in *Simarouba* extracts and fractions exert their effects, in part, by adhering to the eggshells, thereby inhibiting hatching and larval development. Additionally, the use of propidium iodide staining confirmed that these compounds induced structural damage to eggshells, egg membranes, and cuticles of the hatched larvae. Such findings suggest that *S. berteroana* may disrupt the development of *H. contortus* at multiple life stages, highlighting its potential as a natural alternative for parasite control.

Further investigations should focus on toxicity tests, in vivo anthelmintic assays, and pharmacokinetics in order to confirm the extent to which the anthelmintic effects of extracts and fractions of *S. berteroana* can be achieved. By addressing these factors, *S. berteroana* extracts and fractions can be effectively integrated into sustainable parasite control programs, offering cost-effective solutions to combat anthelmintic resistance in small ruminant production systems.

## Figures and Tables

**Figure 1 vetsci-12-00090-f001:**
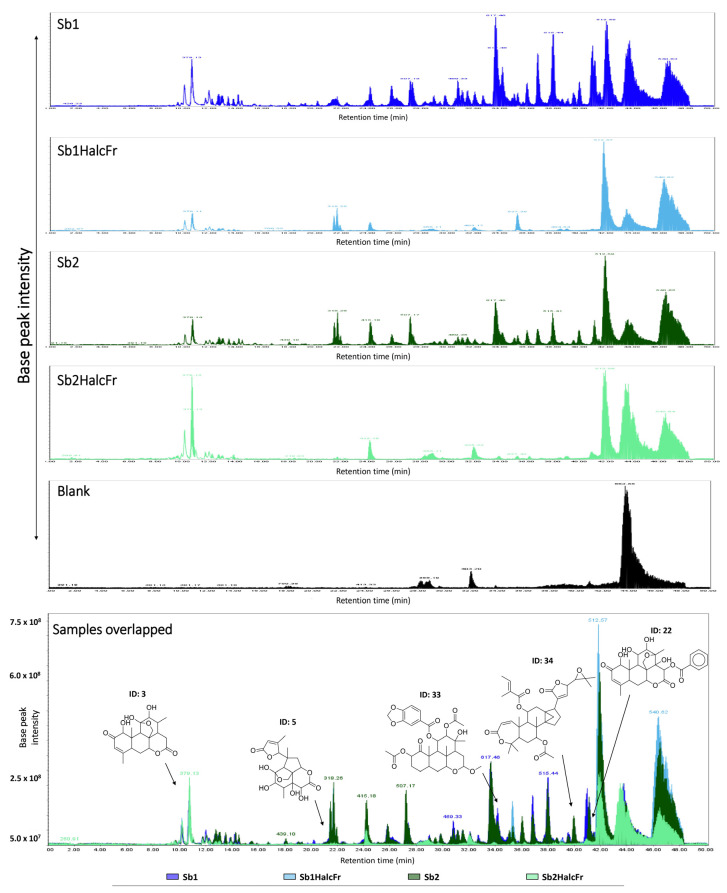
LC-MS analyses of the overlapped base peak ion (BPI) chromatogram of samples (Sb1: crude extract, Sb1Halc: hydroalcoholic fraction from the 1st collection in February 2020, Sb2: crude extract, and Sb2Halc: hydroalcoholic fraction from the 2nd collection in June 2021) in the positive ionization mode with chemical structures of the putatively annotated metabolites. The numbers of the compound are shown in Table 1 (final ID).

**Figure 2 vetsci-12-00090-f002:**
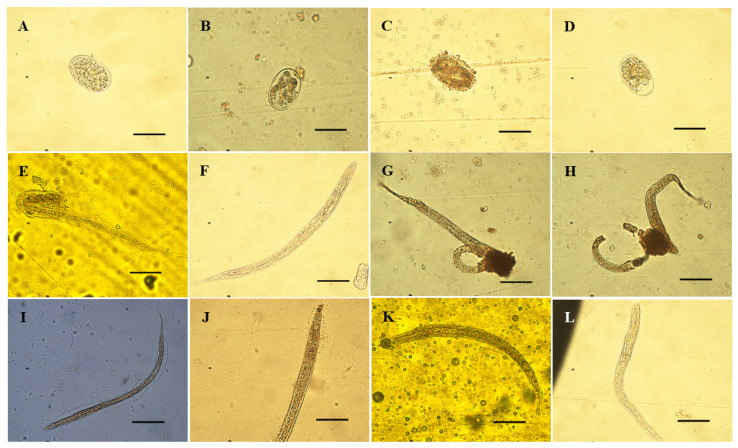
Eggs and 1st-stage larvae of *H. contortus* treated with *S. berteroana iso*-butanol fraction at a concentration of 0.5 mg/mL (**B**,**G**,**H**,**I**), hydroalcoholic fraction at 2.5 mg/mL (**C**) and at 1 mg/mL (**E**,**J**), hexane fraction at 8 mg/mL (**K**), and negative control with 0.5% DMSO in distilled water (**A**,**F**) and thiabendazole at 0.1 mg/mL (**D**,**L**). The black bars in the lower-right margin of the images represent a length of 50 μm.

**Figure 3 vetsci-12-00090-f003:**
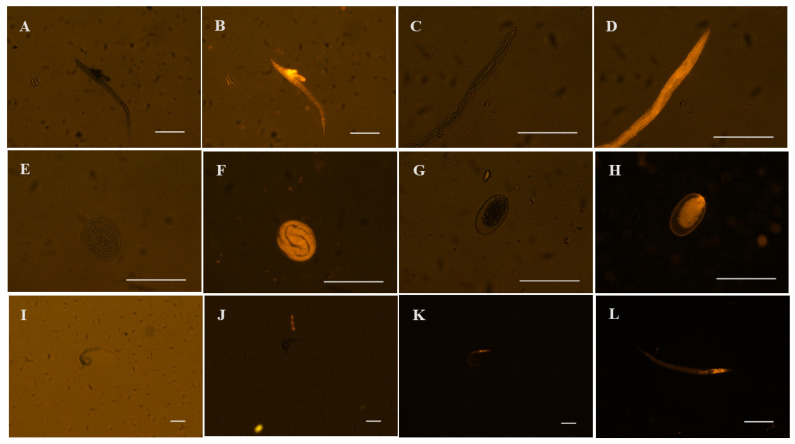
Eggs and 1st-stage larvae of *H. contortus* treated with *S. berteroana* crude extract and hydroalcoholic fraction with and without propidium iodide (**A**–**F**), while (**G**–**L**) were treated with 0.5% DMSO ((**G**,**H**) show an inviable egg and (**I**–**L**) live at the 1st-stage larvae). The white bars in the lower-right margin of the images represent a length of 100 μm.

**Figure 4 vetsci-12-00090-f004:**
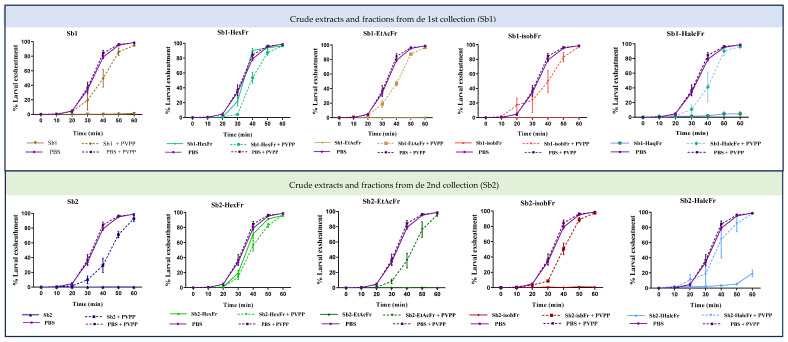
Percentage of exsheathment inhibition (mean and standard error) of 3rd-stage *H. contortus* larvae in contact with the crude extract and fractions of *S. berteroana* at 300 μg/mL. Sb1 and Sb2: *S. berteroana* crude extract derived from Sb1 and Sb2, respectively; the hexane fractions (Sb1-HexFr and Sb2-HexFr), ethyl acetate (Sb1-EtAcFr and Sb2-EtAcFr), *iso*-butanol (Sb1-isobFr and Sb2-isobFr), and hydroalcoholic fractions (Sb1-HalcFr and Sb2-HalcFr) were obtained from each crude extract of *S. berteroana*; PVPP: polyvinylpolypyrrolidone; and PBS: phosphate-buffered solution.

**Table 1 vetsci-12-00090-t001:** Values of total phenolic, tannin, and flavonoid content (mean ± standard deviation) expressed in mg/g of crude extracts and fractions of *Simarouba berteroana*.

*S. berteroana* Extract/Fraction	Crude Extract	HexFr	EtAcFr	isobFr	HalcFr
Phenolics (mg GAE/g)					
Sb1	255.47 ± 22.6 ^b^	-	333.33 ± 34.1 ^ba^	354.33 ± 31.8 ^a^	288.71 ± 9.5 ^abA^
Sb2	276.47 ± 44.7 ^bc^	57.74 ± 61.9 ^d^	354.33 ± 4.6 ^ab^	374.45 ± 21.4 ^a^	189.85 ± 11.8 ^cB^
Tannins (mg TAE/g)					
Sb1	422.04 ± 51.6 ^aA^	-	545.94 ± 29.2 ^b^	528.5 ± 17.0 ^b^	456.95 ± 28.5 ^aB^
Sb2	495.8 ± 26.9 ^aB^	15.45 ± 4.2 ^c^	500.0 ± 31.3 ^a^	517.92 ± 26.0 ^a^	324.19 ± 17.2 ^bA^
Flavonoids (mg QE/g)					
Sb1	22.73 ± 1.1 ^bB^	-	30.88 ± 1.3 ^aA^	20.86 ± 0.6 ^b^	12.46 ± 2.0 ^c^
Sb2	27.49 ± 1.9 ^aA^	-	21.55 ± 2.3 ^bB^	20.80 ± 0.9 ^b^	8.61 ± 1.8 ^c^

mg GAE/g: mg of gallic acid equivalents per gram, mg TAE/g: mg of tannic acid equivalents per gram, mg QE/g: mg of quercetin equivalents per gram of extract/fraction. Different uppercase letters indicate significant differences between columns and different lowercase letters indicate significant differences between rows (*p* < 0.05). Sb1: crude extract of *S. berteroana* collected in February 2020 (1st collection); Sb2: crude extract of *S. berteroana* collected in June 2021 (2nd collection); Fr: extract fraction; HexFr: hexane fraction; EtAcFr: ethyl acetate fraction; isobFr: *iso*-butanol fraction; and HalcFr: hydroalcoholic fraction.

**Table 2 vetsci-12-00090-t002:** LC-MS analyses of the main metabolites detected in the crude extracts and hydroalcoholic fractions of *S. berteroana* (1st and 2nd collections).

Tentative Compound Annotation	Final ID ^a^	RT ^b^	M + H ^c^	Mol. Weight ^d^	MF ^e^	Natural Product Class	Peak Area ^f^ (×10^5^) in Each Extract/Fraction			
							Sb1	Sb1-HalcFr	Sb2	Sb2-HalcFr
Glaucarubol (2-ketone, 15-deoxy)	3	10.8	379	378.1	C_20_H_26_O_7_	Quassinoids	173.0	6.97	197.0	326.0
Dihydroxy Ailanquassin A	5	21.3	397	396.2	C_19_H_24_O_9_	Quassinoids	53.2	0.00	5.01	0.00
Brucein K or Brucein E	8	41.0	414	412.6	C_20_H_28_O_9_	Quinolone quassinoids (terpenoid quinones)	2.31	11.7	59.8	2.25
Tirucalla-7,24-dien-3-one (stigmasta-7,22-dien-3)	9	32.1	425	424.2	C_30_H_48_O	Phytosterol lipids	39.7	82.6	61.9	19.7
Glaucarubol-13,18-didehydro-2-ketone-(2-hydroxy-2-methylbutanoyl)	17	38.2	493	492.4	C_25_H_32_O_10_	Quassinoids	105.0	0.00	19.4	0.00
Glaucarubinon or Glaucarubol-2-ketone (2-hydroxy-2-methylbutanoyl)	18	14.1	495	494.2	C_25_H_34_O_10_	Quassinoids	3.31	17.2	2.9	16.8
Brucein E (2-ketone, 15-benzoyl)	22	37.2	515	514.4	C_27_H_30_O_10_	Quinolone quassinoids (terpenoid quinones)	59.4	0.00	15.8	0.00
Glaucarubol (15-glucopyranoside)	28	33.8	559	558.4	C_26_H_38_O_13_	Glycosylated quassinoids	61.9	0.00	239.0	0.00
Javanicin D	33	34.6	617	616.5	C_32_H_40_O_12_	Naphthoquinones	10.1	0.00	130.0	0.00
Simaroubin B	34	35.3	637	636.4	C_37_H_48_O_9_	Terpenoid quinones	50.5	657.0	38.6	31.6

^a^ ID: annotations by dnp.chemnetbase.com (accessed on 15 December 2024); ^b^ RT: retention time; ^c^ M + H: mass of the ionized protonated molecule; ^d^ molecular weight, ^e^ MF: molecular formula; ^f^ peak intensity: peak area of *m*/*z* at its respective retention time; Sb1 and Sb1-HalcFr: crude extract and the hydroalcoholic fraction of *S. berteroana* (1st collection), respectively; Sb2 and Sb2-HalcFr: crude extract and the hydroalcoholic fraction of *S. berteroana* (1st collection), respectively.

**Table 3 vetsci-12-00090-t003:** Percentage of egg hatch inhibition (mean and standard deviation) of extracts and fractions of *Simarouba berteroana* from the 1st collection (Sb1) against *Haemonchus contortus* Kokstad isolate.

Concentration (mg/mL)	Sb1	HexFr	EtAcFr	IsobFr	HalcFr
16	-	14.43 ± 5.8 ^b^	-	-	-
8	-	7.88 ± 1.05 ^c^	-	-	-
4	100.0 ^aA^	3.11 ± 1.7 ^cdB^	100.0 ^aA^	100.0 ^aA^	100.0 ^aA^
3	100.0 ^a^	-.	100.0 ^a^	100.0 ^a^	100.0 ^a^
2.5	98.72 ± 1.1 ^a^	-	98.32 ± 1.9 ^a^	100.0 ^a^	-
2	76.57 ± 3.2 ^bB^	2.15 ± 1.02 ^dD^	76.25 ± 3.9 ^bBC^	71.91 ± 4.1 ^bC^	100.0 ^aA^
1.5	35.33 ± 2.5 ^cC^	-	33.14 ± 2.6 ^cC^	44.98 ± 3.2 ^cB^	98.51 ± 1.4 ^aA^
1	23.26 ± 2.8 ^dB^	1.45 ± 1.4 ^dC^	25.23 ± 2.7 ^dB^	35.76 ± 3.1 ^dA^	34.77 ± 3.3 ^bA^
0.5	14.74 ± 2.7 ^eA^	1.74 ± 0.3 ^dC^	9.53 ± 1.9 ^eB^	15.27 ± 2.8 ^eA^	18.04 ± 1.8 ^cA^
0.25	9.36 ± 2.1 ^fAB^	-	4.74 ± 1.3 ^fB^	8.36 ± 1.4 ^fAB^	10.67 ± 2.6 ^dA^
0.125	-	-	-	-	7.24 ± 2.4 ^d^
0.5% DMSO	2.78 ± 0.8 ^g^	2.78 ± 0.8 ^d^	2.78 ± 0.8 ^g^	2.78 ± 0.8 ^g^	2.78 ± 0.8 ^e^
Thiabendazole (0.1 mg/mL)	98.49 ± 0.8 ^a^	98.49 ± 0.8 ^a^	98.49 ± 0.8 ^a^	98.49 ± 0.8 ^a^	98.49 ± 0.8 ^a^

Sb1: crude extract of *S. berteroana* collected in February 2020 (1st collection); HexFr: hexane fraction; EtAcFr: ethyl acetate fraction; isobFr: *iso*-butanol fraction; HalcFr: hydroalcoholic fraction; DMSO: dimethylsulfoxide. Different lowercase letters indicate a statistical difference (*p* < 0.05) between rows, and different uppercase letters indicate a statistical difference (*p* < 0.05) between columns, as determined by the Tukey test.

**Table 4 vetsci-12-00090-t004:** Percentage of egg hatch inhibition (mean and standard deviation) of extracts and fractions of *Simarouba berteroana* from the 2nd collection (Sb2) against *Haemonchus contortus* Kokstad isolate.

Concentration (mg/mL)	Sb2	HexFr	EtAcFr	IsobFr	HalcFr
16	-	19.78 ± 2.4 ^b^	-	-	-
8	-	12.68 ± 1.7 ^c^	-	-	-
4	100.0 ^aA^	10.96 ± 1.9 ^cdB^	100.0 ^aA^	100.0 ^aA^	100.0 ^aA^
3	100.0 ^a^	-	100.0 ^a^	100.0 ^a^	100.0 ^a^
2.5	97.62 ± 2.1 ^aA^	-	100.0 ^aA^	100.0 ^aA^	83.79 ± 8.0 ^bB^
2	70.15 ± 6.7 ^bB^	10.54 ± 0.5 ^cdD^	99.18 ± 0.8 ^aA^	100.0 ^aA^	60.73 ± 2.6 ^cC^
1.5	41.08 ± 3.7 ^cB^	-	37.39 ± 4.1 ^bB^	55.08 ± 2.8 ^bA^	35.53 ± 3.8 ^dB^
1	19.99 ± 2.6 ^dC^	6.96 ± 3.1 ^dD^	26.66 ± 3.4 ^cB^	37.33 ± 3.5 ^cA^	21.37 ± 4.2 ^eBC^
0.5	10.11 ± 2.0 ^e^	6.33 ± 1.5 ^de^	10.04 ± 1.9 ^d^	11.52 ± 1.4 ^d^	11.33 ± 2.8 ^f^
0.25	5.29 ± 1.3 ^f^	-	5.67 ± 1.1 ^df^	6.21 ± 1.8 ^de^	5.24 ± 1.4 ^g^
0.5% DMSO	2.61 ± 1.3 ^f^	2.61 ± 1.3 ^e^	2.61 ± 1.3 ^f^	2.61 ± 1.3 ^e^	2.61 ± 1.3 ^g^
Thiabendazole (0.1 mg/mL)	98.28 ± 1.0 ^a^	98.28 ± 1.0 ^a^	98.28 ± 1.0 ^a^	98.28 ± 1.0 ^a^	98.28 ± 1.0 ^a^

Sb2: crude extract of *S. berteroana* collected in June 2021 (2nd collection); HexFr: hexane fraction; EtAcFr: ethyl acetate fraction; isobFr: *iso*-butanol fraction; HalcFr: hydroalcoholic fraction; DMSO: dimethylsulfoxide. Different lowercase letters indicate a statistical difference (*p* < 0.05) between rows, and different uppercase letters indicate a statistical difference (*p* < 0.05) between columns, as shown by the Tukey test.

**Table 5 vetsci-12-00090-t005:** Effective concentration (mean and 95% interval confidence) of extracts and fractions of *S. berteroana* (1st and 2nd collections) for inhibiting 50 and 90% of the egg hatch of *Haemonchus contortus* Kokstad isolate.

Extract/ Fraction	IC_50_ mg/mL		IC_90_ mg/mL	
	Sb1	Sb2	Sb1	Sb2
Crude	1.465 (1.14–1.87) ^a^	1.524 (1.34–1.72) ^a^	2.443 (2.00–3.48) ^a^	2.377 (2.12–2.76) ^a^
HexFr	30.661 (23.75–46.83) ^bA^	37.439 (25.56–82.8) ^bB^	49.167 (37.0–72.54) ^bA^	70.411 (46.44–163.36) ^bB^
EtAcFr	1.515 (1.25–1.82) ^a^	1.338 (0.94–1.83) ^a^	2.392 (2.04–3.09) ^a^	2.057 (1.64–3.35) ^b^
isobFr	1.382 (1.1–1.7) ^a^	1.196 (0.93–1.49) ^a^	2.388 (2.0–3.2) ^a^	1.897 (1.58–2.56) ^b^
HalcFr	0.925 (0.56–1.3) ^a^	1.662 (1.48–1.85) ^a^	1.517 (1.18–2.4) ^a^	2.708 (2.44–3.1) ^b^

Sb1 and Sb2: *Simarouba* from the 1st (February 2020) and 2nd (June 2021) collections, respectively; Crude: crude extract; HexFr: hexane fraction; EtAcFr: ethyl acetate fraction; isobFr: *iso*-butanol fraction; HalcFr: hydroalcoholic fraction; and IC_50_ and IC_90_: inhibitory concentration of 50% and 90%, respectively. Different lowercase letters indicate a statistical difference (*p* < 0.05) between rows and different uppercase letters indicate a statistical difference (*p* < 0.05) between columns in the Tukey test.

## Data Availability

The original contributions presented in the study are included in the article material. Some of the data supporting the results of this study are available in the Appendix A to this article. Further inquiries can be directed to the corresponding author.

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
