# Peer review of "Simarouba berteroana Krug & Urb. Extracts and Fractions Possess Anthelmintic Activity Against Eggs and Larvae of Multidrug-Resistant Haemonchus contortus"

_vetsci, 2025, doi:10.3390/vetsci12020090_

Round 1

Reviewer 1 Report

Comments and Suggestions for Authors

Very nice work done. I have only one comment.

Can you please add information on the potential in vivo use of this substance?

Perhaps some similar in vivo experiments with the same of similar substances from the literature. I think this is vert important. Thanks in advance.

Author Response

Comments 1: [Very nice work done. I have only one comment.

Can you please add information on the potential in vivo use of this substance?

Perhaps some similar in vivo experiments with the same of similar substances from the literature. I think this is very important. Thanks in advance.]

Response 1: [Thank you for your suggestion. Regrettably, there is a scarcity of studies evaluating the in vivo anthelmintic efficacy of plants belonging to the Simarouba genus and the Simaroubaceae family. Nevertheless, some research has investigated the larvicidal activity of extracts from certain Simaroubaceae plants against various nematode species. While these parasitological tests do provide promising results, immunological, histopathological and pharmacokinetic assays are necessary to truly evaluate the in vivo potential of plant extracts.  Taking your comment into account, we have included two sentences in the discussion, which can now be found in lines 766–770.]   

In the attachment, you will also find our responses to your comments.

Additional changes:

  • In line 581, we removed the words “except for” and replaced them with “with” to avoid redundancy.

Reviewer 2 Report

Comments and Suggestions for Authors

This study “Simarouba berteroana Krug & Urb. Extracts and Fractions Possess Anthelmintic Activity Against Eggs and Larvae of Multidrug-Resistant Haemonchus contortus” explores the anthelmintic properties of Simarouba berteroana extracts and their fractions against a multidrug-resistant isolate of Haemonchus contortus. The paper is well-structured, with clear objectives, comprehensive methodologies, and detailed analyses. The findings contribute to the search for alternative anthelmintic treatments in veterinary science. I have only some minor comments for the improvement the paper. As described by author, it is well-documented that tannin good anthelmintic activity against nematodes. However, condensed tannin can pose anti-nutritional problems to ruminants.

Minor Comments:

Line 224- 0.5%DMSO was used for negative control. Did you use DMSO for dilution of plant extracts? It is not clear.

Line 355- Juny 2021 means June or July?

In Table 1- ensure to adjust table column (mean ± standard deviation).

In discussion- You brought up the possibility that tannin (and flavonoids) was responsible for the activities against eggs and larvae of parasites. However, you should be aware that condensed tannin can be anti-nutritional problems to ruminants.

Author Response

Comment 1: Line 224- 0.5%DMSO was used for negative control. Did you use DMSO for dilution of plant extracts? It is not clear.

Response: Thank you for the observation. DMSO was in fact used for the dilution of the extracts and their fractions in distilled water for the egg hatch test. We have added this information in the Methods section (line 223, page 5).

Comment 2: Line 355- Juny 2021 means June or July?

Response: We apologize for the error; the correct date is June 2021. Thank you for bringing this to our attention. In the revised version of the manuscript, we have already made the correction (line 355).

Comment 3: In Table 1- ensure to adjust table column (mean ± standard deviation).

Response: We are grateful for the observation. In response, we have adjusted the dimensions of the columns in Table 1 (page 7), ensuring uniformity. Additionally, the font size in the cells and headers was reduced from 9 to 8.5 and 10 to 9, respectively, to improve visualization.

Comment 4: In discussion- You brought up the possibility that tannin (and flavonoids) was responsible for the activities against eggs and larvae of parasites. However, you should be aware that condensed tannin can be anti-nutritional problems to ruminants.

Response: You are absolutely correct, as the ability of tannins, particularly condensed tannins, to form complexes with proteins reduces their digestibility and also causes a decrease in feed intake, according to Besharati et al. 2022 and Naumann et al. 2017 (10.3390/molecules27238273 and 10.1590/s1806-92902017001200009). In light of this, we have added a few lines (759 - 765) to the discussion, addressing this point. We also suggest that future studies on the in vivo anthelmintic effect of S. berteroana extracts should evaluate the potential for anti-nutritional effects in small ruminants. Therefore, two new citations were included in the list of references (lines 1016 - 1019).

In the attachment, you will also find our responses to your comments.

Additional changes:

  • In line 581, we removed the words “except for” and replaced them with “with” to avoid redundancy.
